# Heterogeneous Treatment Effect with Trained Kernels of the Nadaraya–Watson Regression

Andrei Konstantinov [†] , Stanislav Kirpichenko [†] and Lev Utkin *,[†]

Institute of Computer Science and Technology, Peter the Great St. Petersburg Polytechnic University, Polytechnicheskaya, 29, 195251 St. Petersburg, Russia; konstantinov_av@spbstu.ru (A.K.); kirpichenko.sr@edu.spbstu.ru (S.K.)

* Correspondence: utkin_lv@spbstu.ru
† These authors contributed equally to this work.

**Abstract:** A new method for estimating the conditional average treatment effect is proposed in this paper. It is called TNW-CATE (the Trainable Nadaraya–Watson regression for CATE) and based on the assumption that the number of controls is rather large and the number of treatments is small. TNW-CATE uses the Nadaraya–Watson regression for predicting outcomes of patients from control and treatment groups. The main idea behind TNW-CATE is to train kernels of the Nadaraya–Watson regression by using a weight sharing neural network of a specific form. The network is trained on controls, and it replaces standard kernels with a set of neural subnetworks with shared parameters such that every subnetwork implements the trainable kernel, but the whole network implements the Nadaraya–Watson estimator. The network memorizes how the feature vectors are located in the feature space. The proposed approach is similar to transfer learning when domains of source and target data are similar, but the tasks are different. Various numerical simulation experiments illustrate TNW-CATE and compare it with the well-known T-learner, S-learner, and X-learner for several types of control and treatment outcome functions. The code of proposed algorithms implementing TNW-CATE is publicly available.

**Keywords:** treatment effect; Nadaraya–Watson regression; neural network; shared weights; meta-learner; regression





## 1. Introduction

The efficient treatment for a patient with her/his clinical and other characteristics [1,2] can be regarded as an important goal of the real personalized medicine. The problem is that patients differ not only in their background characteristics, but also in how they respond to a particular treatment [3]. Therefore, we need to draw inferences about individual-level treatment effects, as opposed to inferring treatment effects on average across a set of patients [4,5]. The goal of personalized medicine, which takes into account the difference between patients, can be achieved by using machine learning methods due to the increasing amount of available electronic health records which are a basis to develop accurate models. To estimate the treatment effect, patients are divided into two groups called treatment and control, and then patients from the different groups are compared. One of the popular measures of efficient treatment used in machine learning models is the average treatment effect (ATE) [6], which is estimated on the basis of observed data regarding the mean difference between outcomes of patients from the treatment and control groups. Due to the difference between characteristics of patients and the difference between their responses to a particular treatment, the treatment effect is measured by the conditional average treatment effect (CATE) or the heterogeneous treatment effect (HTE) that is defined as ATE, which is conditional on a patient feature vector [7–10].

Two main problems can be observed when CATE is estimated. The first one is that the control group is usually larger than the treatment group. Hence, we meet the problem of a

small training dataset, which does not allow us to apply many efficient machine learning methods directly. Aoki and Ester [11] consider an example of the difficulties in collecting corresponding information about patients and to explore the side effects of drugs adopted on certain pediatric cancer treatments for several reasons, including the sensitivity of issues for the families, the rarity of the disease, and the effort required between hospitals and doctors to refer their patients. The authors propose to use a transfer learning approach, implemented by means of a neural network, to estimate treatment effects on small datasets. This is an efficient and interesting approach. However, it may encounter difficulties when training the neural network that models the regression functions for controls and treatments. The same problem has been pointed out in [4,12]. Therefore, we need to develop a similar approach where the neural networks should be rather simple and play a secondary role in the modeling the regression functions.

The second problem is that each patient cannot be simultaneously in the treatment and control groups, i.e., we either observe the patient outcome under treatment or control, but never both [13]. This is a fundamental problem of computing the causal effect. This problem is solved by the explicit or implicit construction of regression functions for the control and treatment patients. For instance, we can train neural networks [14], which predict outcomes for a new patient under condition that this patient belongs to one of the control or treatment groups. However, this method requires one to have large datasets to train the corresponding machine learning models. We thus return to the first problem of the treatment group size.

In addition to the above two problems, there are many difficulties facing machine learning model development concerning noisy data, especially with the high dimension of the patients' health records, etc. [15]. One of the difficulties is a complex data structure. A lot of methods have some assumptions about the model parameters and the data structure. For instance, a Gaussian mixture for outcomes is proposed for use in [11]. In some cases, these assumptions are correct, and they correspond to the real structure of the treatment and control data. However, they can lead to errors in other cases where the data structure is rather complex. Dorie et al. [16] note that the analysis of observations, having a grouped structure, shows that the impact of the treatment exposure will vary across these groups. In this case, most current machine learning approaches ignore these varying effects. Therefore, it is necessary to develop models that take into account these problems.

Many methods for estimating CATE have been proposed and developed due to importance of the problem in medicine and other applied areas [12,17–29]. This is only a small part of all publications which are devoted to the CATE estimation problem solution. Various approaches were used for solving the problem, including the support vector machine [30], tree-based models [15], neural networks [2,13,31,32], and transformers [33–36].

It should be noted that most approaches to estimating CATE are based on constructing regression models for handling the treatment and control groups. However, the problem of the small treatment group motivates us to develop various tricks that at least partially resolve the problem.

We propose a method based on using the Nadaraya–Watson kernel regression [37,38] which is widely applied to machine learning problems. The method is called TNW-CATE (the Trainable Nadaraya–Watson regression for CATE). The Nadaraya–Watson estimator can be seen as a weighted average of outcomes (patient responses) by means of a set of varying weights called attention weights. The attention weights in the Nadaraya–Watson regression are defined through kernels that measure distances between training feature vectors and the target feature vector, i.e., kernels in the Nadaraya–Watson regression conform with relevance of a training feature vector to a target feature vector. If we have a dataset $\{(\mathbf{x}_1, y_1), \ldots, (\mathbf{x}_n, y_n)\}$, where $\mathbf{x}_i \in \mathbb{R}^m$ is a feature vector (key) and $y_i \in \mathbb{R}$ is its

target value or its label (value), then the Nadaraya–Watson estimator for a target feature vector $\mathbf{z} \in \mathbb{R}^m$ (query) can be defined by using weights $\alpha(\mathbf{z}, \mathbf{x}_i)$ as

$$\hat{f}(\mathbf{z}) = \sum_{i=1}^{n} \alpha(\mathbf{z}, \mathbf{x}_i) y_i, \tag{1}$$

where

$$\alpha(\mathbf{z}, \mathbf{x}_i) = \frac{K(\mathbf{z}, \mathbf{x}_i, \gamma)}{\sum_{j=1}^{n} K(\mathbf{z}, \mathbf{x}_j, \gamma)}, \tag{2}$$

$K(\mathbf{z}, \mathbf{x}_i, \gamma)$ is a kernel and $\gamma > 0$ is a bandwidth parameter.

Standard kernels widely used in practice are the Gaussian, uniform, or Epanechnikov kernels [39]. However, the choice of a kernel and its parameters significantly impact on results obtained from the Nadaraya–Watson regression usage. Moreover, the Nadaraya–Watson regression also requires a large number of training examples. Therefore, we propose a quite different way for implementing the Nadaraya–Watson regression. The method is based on the following assumptions and ideas. First, each kernel is represented as a part of a neural network implementing the Nadaraya–Watson regression. In other words, we do not use any predefined standard kernels such as the Gaussian one. Kernels are trained as the weight sharing neural subnetworks. The weight sharing is used to identically compute kernels under the condition that the pair $(\mathbf{z}, \mathbf{x}_i)$ of examples is fed into every subnetwork. The neural network kernels become more flexible and sensitive to a complex location structure of feature vectors. In fact, we propose to replace the definition of weights through the kernels with a set of neural subnetworks with shared parameters (the neural network weights) such that every subnetwork implements the trainable kernel, but the whole network implements the Nadaraya–Watson estimator. At that, the trainable parameters of the kernels are nothing else but the weights of each neural subnetwork. The above implementation of the Nadaraya–Watson regression by means of the neural network leads to an interesting result when the treatment examples are considered as a single example whose "features" are the whole treatment feature vectors.

It should be noted that several authors [40–43] proposed to use the Nadaraya–Watson kernel regression with standard kernels having the bandwidth parameter to construct the CATE estimator. However, they did not propose to learn the kernels. Moreover, they did not propose to use the kernels in the framework of transfer learning where the kernels are trained on controls, but they are used for treatments. The Nadaraya–Watson kernel regression in these works has a relative disadvantage. It requires one to define a certain kernel for computing weights of examples, for example, the Gaussian kernel. We overcome the above difficulty by replacing the standard kernel with a neural network which implements the kernel.

The second assumption is that the feature vector domains of the treatment and control groups are similar. For instance, if some components of the feature vectors from the control group are logarithmically located in the feature space, then the feature vectors from the treatment group have the same tendency to be located in the feature space. Figure 1 illustrates the corresponding location of the feature vectors. Vectors from the control and treatment groups are depicted by small circles and triangles, respectively. It can be seen from Figure 1 that the control examples as well as the treatment ones are located unevenly along the *x*-axis. Many standard regression methods do not take into account this peculiarity. It should be noted that this assumption is often fulfilled because patients are treated after they were in the control group with a particular disease. A treatment is usually studied for patients with a specific disease. Therefore, feature vector domains of patients from the control and treatment groups are close to each other. Another illustration is shown in Figure 2 where the feature vectors are located on spirals, but the spirals have different values of the patient outcomes. Even if we were to use a kernel regression, it would be difficult to find such standard kernels satisfying the training data. However, the assumption of the similarity of domains allows us to train kernels on examples from

the control group because the kernels depend only on the feature vectors. In this case, the network memorizes how the feature vectors are located in the feature space. By using assumption about similarity of the treatment and control domains, we can apply the Nadaraya–Watson regression with the trained kernels to the treatment group changing the patient outcomes. TNW-CATE is similar to the transfer learning [44–46] when domains of source and target data are the same, but tasks are different. Therefore, the abbreviation TNW-CATE can be also read as the transferable kernels of the Nadaraya–Watson regression for the CATE estimating.

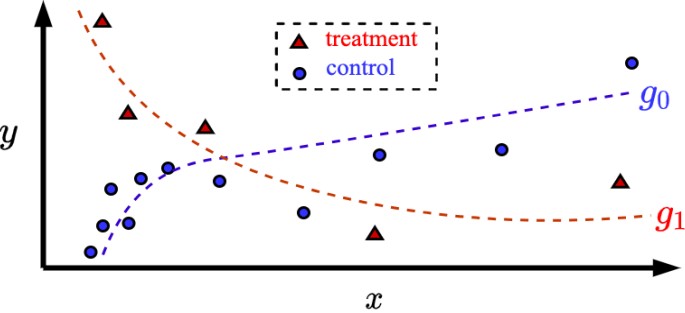

**Figure 1.** Illustration of the logarithmical location of feature vectors corresponding to patients from the treatment group (small triangles) and from the control group (small circles).

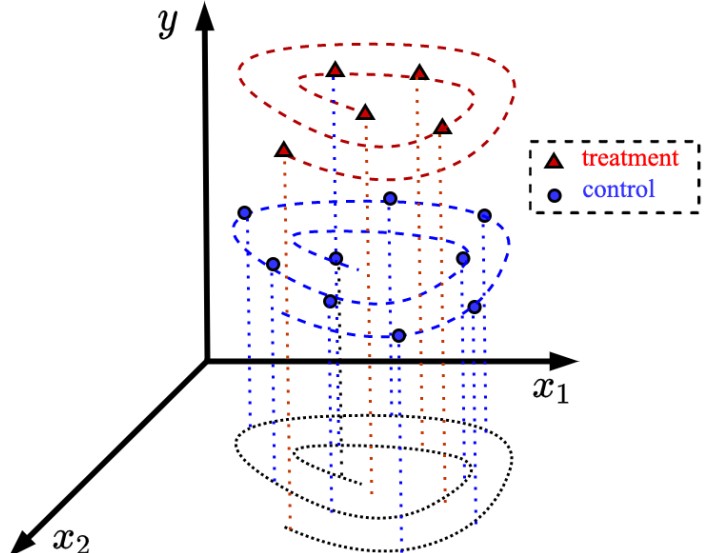

**Figure 2.** Illustration of the location of feature vectors corresponding to patients from the treatment group (small triangles) and from the control group (small circles) on spirals.

Our contributions can be summarized as follows:

1.  We propose to use the Nadaraya–Watson kernel regression which does not rely on specific regression functions and estimates regression values (outputs of controls and treatments) without any assumptions about the functions. The main feature of the model is that kernels of the Nadaraya–Watson regression are implemented as neural networks, i.e., the kernels are trained on the control and treatment data. In contrast to many CATE estimators based on neural networks, the proposed model uses simple neural networks which implement only kernels, but not the regression functions.
2.  The proposed model and the neural network architecture allow us to solve the problem of the small numbers of patients in the treatment group. This is a crucial problem

especially when new treatments and new drugs are tested. In fact, the proposed model can be considered in the framework of the transfer learning when controls can be viewed as source data (in terms of the transfer learning), but the treatments are target data.

3. Neural networks implementing the kernels amplifies the model flexibility. In contrast to the standard kernels, the neural kernels allow us to cope with the possible complex data structure because they are adapted to the structure due to many trainable parameters.

4. A specific algorithm of training the neural kernels is proposed. It trains networks on controls and treatments simultaneously in order to memorize the treatment data structure. We show by means of numerical examples that there is an optimal linear combination of two loss functions corresponding to the controls and treatments.

Various numerical experiments study peculiarities of TNW-CATE and illustrate the outperformance of TNW-CATE in comparison with the well-known meta-models: the T-learner, the S-learner, and the X-learner for several control and treatment output functions. The code of the proposed algorithms can be found at https://github.com/Stasychbr/TNW-CATE (accessed on 19 April 2023).

The paper is organized as follows. Section 2 can be viewed as a review of existing models for estimating CATE. Applications of the Nadaraya–Watson regression in machine learning also are considered in this section. A formal statement of the CATE estimation problem is provided in Section 3. TNW-CATE and the main ideas and algorithms for implementing TNW-CATE are described in Section 4. Numerical experiments illustrating TNW-CATE and comparing it with other models are presented in Section 5. Concluding remarks are provided in Section 6.

## 2. Related Work

**Estimating CATE**. Computing CATE is a very important problem which can be regarded as a tool for implementing personalized medicine [47]. This fact motivated researchers to develop many efficient approaches solving the problem. An approach based on the Lasso model for estimating CATE was proposed by Jeng et al. [48]. Several interesting approaches using the SVM model were presented in [30,49]. A "honest" model for computing CATE was proposed by Athey and Imbens [18]. According to the model, the training set is split into two subsets such that the first one is used to construct the partition of the data into subpopulations that differ in the magnitude of their treatment effect, and the second subset is used to estimate treatment effects for each subpopulation. A unified framework for constructing fast tree-growing procedures solving the CATE problem was provided in [50,51]. A modification of the survival causal tree method for estimating the CATE based on censored observational data was proposed in [52]. Xie et al. [53] established the CATE detection problem as a false positive rate control problem, and they discussed in detail the importance of this approach for solving large-scale CATE detection problems. Algorithms for estimating CATE in the context of Bayesian nonparametric inference were studied in [12]. Bayesian additive regression trees, causal forest, and causal boosting were compared under condition of binary outcomes in [15]. An orthogonal random forest as an algorithm that combines orthogonalization with generalized random forests for solving the CATE estimation problem was proposed in [54]. Estimating CATE as the anomaly detection problem was studied in [55]. Many other approaches can also be found in [3,47,56–64].

A set of meta-algorithms or meta-learners, including the T-learner [25], the S-learner [25], the O-learner [65], and the X-learner [25] were investigated and compared in [25].

Neural networks can be regarded as one of the efficient tools for estimating CATE. As a result, many models based on using neural networks have been proposed [2,13,14,31,32,66–73].

Transformer-based architectures using attention operations [74] were also applied to solving the CATE estimating problem [33–36]. Ideas of applying the transfer learning technique to CATE estimation were considered in [11,13,75]. Ideas of using the Nadaraya–

Watson kernel regression in CATE estimation were studied in [76,77] where it was shown that the Nadaraya–Watson regression can be used for CATE estimation. However, the small number of training examples in the treatment group does not allow us to efficiently apply this approach. Therefore, we aim to overcome this problem by introducing a neural network of a special architecture, which implements the trainable kernels in the Nadaraya–Watson regression.

**The Nadaraya–Watson regression in machine learning**. There are several machine learning approaches based on applying the Nadaraya–Watson regression [78–84]. Properties of the boosting with kernel regression estimates as weak learners were studied in [85]. A metric learning model with the Nadaraya–Watson kernel regression was proposed in [86]. The high-dimensional nonparametric regression models were considered in [87]. Models taking into account the available correlated errors were proposed in [88]. An interesting work discussing a problem of embedding the Nadaraya–Watson regression into the neural network as a novel trainable CNN layer was presented in [89]. Applied machine learning problems solved by using the Nadaraya–Watson regression were considered in [90]. A method of approximation using the kernel functions made from only the sample points in the neighborhood of input values to simplify the Nadaraya–Watson estimator is proposed in [91]. An interesting application of the Nadaraya–Watson regression to improving the local explanation method SHAP is presented in [92] where the authors find that the Nadaraya–Watson estimator can be expressed as a self-normalized importance sampling estimator. An explanation of how the Nadaraya–Watson regression can be regarded as a basis for understanding the attention mechanism from the statistics point of view can be found in [74,93].

In contrast to the above works, we pursue two goals. The first one is to show how kernels in the Nadaraya–Watson regression can be implemented and trained as neural networks of a special form. The second goal is to apply the whole neural network implementing the Nadaraya–Watson regression to the problem of estimating CATE.

### 3. A Formal Problem Statement

Suppose that the control group of patients is represented as a set of $c$ examples of the form $\mathcal{C} = \{(\mathbf{x}_1, y_1), \ldots, (\mathbf{x}_c, y_c)\}$, where $\mathbf{x}_i = (x_{i1}, \ldots, x_{id}) \in \mathbb{R}^d$ is the $d$-dimensional feature vector for the $i$-th patient from the control group; $y_i \in \mathbb{R}$ is the $i$-th observed outcome, for example, time to death of the $i$-th patient from the control group or the blood sugar level of this patient. The similar notations can be introduced for the treatment group containing $t$ patients, namely, $\mathcal{T} = \{(\mathbf{z}_1, h_1), \ldots, (\mathbf{z}_t, h_t)\}$. Here $\mathbf{z}_i = (z_{i1}, \ldots, z_{id}) \in \mathbb{R}^d$ and $h_i \in \mathbb{R}$ are the feature vector and the outcome of the $i$-th patient from the treatment group, respectively. We will also use the notation of the treatment assignment indicator $T_i \in \{0, 1\}$ where $T_i = 0$ ($T_i = 1$) corresponds to the control (treatment) group.

Let $Y$ and $H$ denote the potential outcomes of a patient if $T = 0$ and $T = 1$ for the patient, respectively. Let $\mathbf{X}$ be the random feature vector from $\mathbb{R}^d$. The treatment effect for a new patient with the feature vector $\mathbf{x}$, which shows how the treatment is useful and efficient, is estimated by the Individual Treatment Effect (ITE) defined as $H - Y$. Since the ITE cannot be observed, then the treatment effect is estimated by means of CATE which is defined as the expected difference between two potential outcomes as follows [94]:

$$\tau(\mathbf{x}) = \mathbb{E}[H - Y \mid \mathbf{X} = \mathbf{x}]. \tag{3}$$

The fundamental problem of computing CATE is that, for each patient in the training dataset, we can observe only one of outcomes $y$ or $h$. To overcome this problem, the important assumption of unconfoundedness [95] is used to allow the untreated units to be used to construct an unbiased counterfactual for the treatment group [76]. According to the assumption of unconfoundedness, the treatment assignment $T$ is independent of the potential outcomes for $Y$ or H conditional on $\mathbf{x} = \mathbf{z}$, respectively, which can be written as

$$T \perp \{Y, H\} \mid \mathbf{x}. \tag{4}$$

Another assumption called the overlap assumption regards the joint distribution of treatments and covariates. According to this assumption, there is a positive probability of being both treated and untreated for each value of **x**. It is of the form:

$$0 < \Pr\{T = 1 \mid \mathbf{x}\} < 1. \tag{5}$$

If we are to accept these assumptions, then CATE can be represented as follows:

$$\tau(\mathbf{x}) = \mathbb{E}[H \mid \mathbf{X} = \mathbf{x}] - \mathbb{E}[Y \mid \mathbf{X} = \mathbf{z}]. \tag{6}$$

The motivation behind unconfoundedness is that nearby observations in the feature space can be treated as having come from a randomized experiment [10].

If we suppose that outcomes of patients from the control and treatment groups are expressed through the functions $g_0$ and $g_1$ of the feature vectors $X$, then the corresponding regression functions can be written as

$$y = g_0(\mathbf{x}) + \varepsilon, \ \mathbf{x} \in \mathcal{C}, \tag{7}$$

$$h = g_1(\mathbf{z}) + \varepsilon, \ \mathbf{z} \in \mathcal{T}. \tag{8}$$

Here, $\varepsilon$ is a Gaussian noise variable such that $\mathbb{E}[\varepsilon] = 0$. Hence, there holds under condition $\mathbf{x} = \mathbf{z}$

$$\tau(\mathbf{x}) = g_1(\mathbf{x}) - g_0(\mathbf{x}). \tag{9}$$

An example illustrating sets of controls (small circles), treatments (small triangles), and the corresponding unknown function $g_0$ and $g_1$ are shown in Figure 1.

## 4. The TNW-CATE Description

It has been mentioned that the main idea behind TNW-CATE is to replace the Nadaraya–Watson regression with the neural network of a specific form. The whole network consists of two main parts. The first part implements the Nadaraya–Watson regression for training the control function $g_0(\mathbf{x})$. In turn, it consists of $n$ identical subnetworks such that each subnetwork implements the attention weight $\alpha(\mathbf{x}, \mathbf{x}_i)$ or the kernel $K(\mathbf{x}, \mathbf{x}_i)$ of the Nadaraya–Watson regression. Therefore, the input of each subnetwork is two vectors $\mathbf{x}$ and $\mathbf{x}_i$, i.e., two vectors $\mathbf{x}$ and $\mathbf{x}_i$ are fed to each subnetwork. The whole network consisting of $n$ identical subnetworks and implementing the Nadaraya–Watson regression for training the control function will be called the control network. In order to train the control network, for every vector $\mathbf{x}_i$, $i = 1, \ldots, c$, $N$ subsets of size $n$ are randomly selected from the control set $\mathcal{C}$ without example $(\mathbf{x}_i, y_i)$. The subsets can be regarded as $N$ examples for training the network. Hence, the control network is trained on $N \cdot c$ examples of size $2d \cdot n$. If we have a feature vector $\mathbf{x}$ for estimating $\tilde{y}$,, i.e., for estimating function $g_0(\mathbf{x})$, then it is fed to each trained subnetworks jointly with each $\mathbf{x}_i$, $i = 1, \ldots, c$, from the training set. In this case, the trained subnetworks or kernels of the Nadaraya–Watson regression are used to estimate $\tilde{y}$. The number of subnetworks for testing is equal to the number of training examples $n$ in every subset. Since the trained subnetworks are identical and have the same weights (parameters), their number can be arbitrary. In fact, a single subnetwork can be used in practice, but its output depends on the pair of vectors $\mathbf{x}$ and $\mathbf{x}_i$.

The same network called the treatment network is constructed for the treatment group. In contrast to the control network, it consists of $m$ subnetworks with inputs in the form of pairs $(\mathbf{z}, \mathbf{z}_i)$. In the same way, $M$ subsets of size $m$ are randomly selected from the training set of treatments without the example $(\mathbf{z}_j, h_j)$, $j = 1, \ldots, t$. The treatment network is trained on $M \cdot t$ examples of size $2d \cdot m$. After training, the treatment network allows us to estimate $\tilde{h}$ as function $g_1(\mathbf{z})$. If $\mathbf{z} = \mathbf{x}$, then we obtain an estimate of CATE $\tau(\mathbf{x})$ or $\tau(\mathbf{z})$ as the difference between estimates $\tilde{h}$ and $\tilde{y}$ obtained by using the treatment and control neural networks. It is important to point out that the control and treatment networks are jointly trained by using the loss function defined below.

Let us formally describe TNW-CATE in detail. Consider the control group of patients $\mathcal{C} = \{(\mathbf{x}_1, y_1), \ldots, (\mathbf{x}_c, y_c)\}$. For every $i$ from set $\{1, \ldots, c\}$, we define $N$ subsets $\mathcal{C}_{i,r}$, $r = 1, \ldots, N$, consisting of $n$ examples randomly selected from $\mathcal{C} \setminus (\mathbf{x}_i, y_i)$ as:

$$\mathcal{C}_{i,r} = \{(\mathbf{x}_1^{(r)}, y_1^{(r)}), \ldots, (\mathbf{x}_n^{(r)}, y_n^{(r)})\},$$
$$r = 1, \ldots, N, \tag{10}$$

where $\mathbf{x}_j^{(r)}$ is a randomly selected vector of features from the set $\{\mathbf{x}_1, \ldots, \mathbf{x}_c\} \setminus \mathbf{x}_i$, which forms $\mathcal{C}_{i,r}$; $y_j^{(r)}$ is the corresponding outcome.

Each subset $\mathcal{C}_{i,r}$ jointly with $(\mathbf{x}_i, y_i)$ forms a training example for the control network as follows:

$$\mathbf{a}_i^{(r)} = \left(\mathbf{x}_1^{(r)}, \ldots, \mathbf{x}_n^{(r)}, \mathbf{x}_i, y_1^{(r)}, \ldots, y_n^{(r)}, y_i\right),$$
$$i = 1, \ldots, c, \ r = 1, \ldots, N. \tag{11}$$

If we feed this example to the control network, then we expect to obtain some approximation $\tilde{y}_i^{(r)}$ of $y_i$. The number of the above examples for training is $N \cdot c$.

Let us consider the treatment group of patients $\mathcal{T} = \{(\mathbf{z}_1, h_1), \ldots, (\mathbf{z}_t, h_t)\}$ now. For every $j$ from set $\{1, \ldots, t\}$, we define $M$ subsets $\mathcal{T}_{j,s}$ and $s = 1, \ldots, M$, consisting of $m$ examples randomly selected from $\mathcal{T} \setminus (\mathbf{z}_j, h_j)$ as:

$$\mathcal{T}_{j,s} = \{(\mathbf{z}_1^{(s)}, h_1^{(s)}), \ldots, (\mathbf{z}_m^{(s)}, h_m^{(s)})\},$$
$$s = 1, \ldots, M, \tag{12}$$

where $\mathbf{z}_l^{(s)}$ is a randomly selected vector of features from the set $\{\mathbf{z}_1, \ldots, \mathbf{z}_t\} \setminus \mathbf{z}_j$, which forms $\mathcal{T}_{j,s}$; $h_j^{(s)}$ is the corresponding outcome.

Each subset $\mathcal{T}_{j,s}$ jointly with $(\mathbf{z}_j, h_j)$ forms a training example for the control network as follows:

$$\mathbf{b}_j^{(s)} = \left(\mathbf{z}_1^{(s)}, \ldots, \mathbf{z}_m^{(s)}, \mathbf{z}_j, h_1^{(s)}, \ldots, h_m^{(s)}, h_j\right),$$
$$j = 1, \ldots, t, \ s = 1, \ldots, M. \tag{13}$$

If we feed this example to the treatment network, then we expect to obtain some approximation $\tilde{h}_j^{(s)}$ of $h_j$. Indices $r$ and $s$ are used to distinguish subsets of controls and treatments.

The architecture of the joint neural network consisting of the control and treatment networks is shown in Figure 3. One can see from Figure 3 that normalized outputs $\alpha_{i,j}^{(r)}$ and $\delta_{j,l}^{(s)}$ of the subnetworks in the control and treatment networks are multiplied by $y_j^{(r)}$ and $h_l^{(s)}$, respectively, and then the obtained results are summed. Here $\alpha_{i,1}^{(r)} + \ldots + \alpha_{i,n}^{(r)} = 1$ and $\delta_{j,1}^{(s)} + \ldots + \delta_{j,m}^{(s)} = 1$. It should be noted again that the control and treatment networks have the same parameters (weights). Every network implements the Nadaraya–Watson regression with this architecture, i.e.,

$$\tilde{y}_i^{(r)} = g_0(\mathbf{x}_i) = \sum_{j=1}^n \alpha_{i,j}^{(r)} y_j^{(k)},$$

$$\tilde{h}_j^{(s)} = g_1(\mathbf{z}_j) = \sum_{j=1}^n \delta_{i,j}^{(s)} h_j^{(s)}, \tag{14}$$

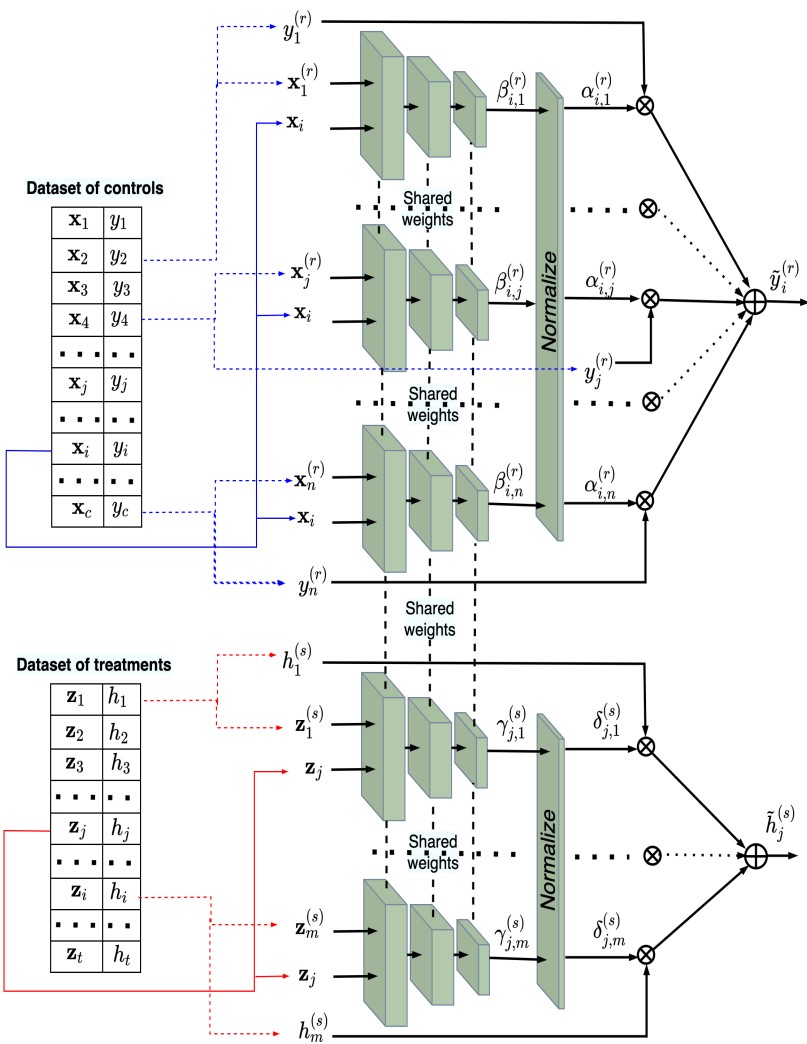

**Figure 3.** The weight sharing neural network consisting of *n* subnetworks for training on the controls and *m* subnetworks for training on the treatments, which implement two models of the Nadaraya–Watson regression.

Here,

$$\alpha_{i,j}^{(r)} = \alpha\left(\mathbf{x}_i, \mathbf{x}_j^{(r)}\right) = \frac{K\left(\mathbf{x}_i, \mathbf{x}_j^{(r)}\right)}{\sum_{k=1}^{n} K\left(\mathbf{x}_i, \mathbf{x}_k^{(r)}\right)}, \tag{15}$$

$$\delta_{i,j}^{(s)} = \delta\left(\mathbf{z}_j, \mathbf{z}_i^{(s)}\right) = \frac{K\left(\mathbf{z}_j, \mathbf{z}_i^{(s)}\right)}{\sum_{k=1}^{m} K\left(\mathbf{z}_j, \mathbf{z}_k^{(s)}\right)}. \tag{16}$$

If we consider the whole neural network, then the training examples are of the form:

$$\left(\mathbf{a}_i^{(r)}, \mathbf{b}_j^{(s)}\right), \ i = 1, \dots, c, \ j = 1, \dots, t,$$

$$r = 1, \dots, N, \ s = 1, \dots, M. \tag{17}$$

The standard expected $L_2$ loss function for training the whole network is of the form:

$$L = \frac{1}{N \cdot c} \sum_{r=1}^{N} \sum_{i=1}^{c} \left( \tilde{y}_i^{(r)} - y_i \right)^2$$
$$+ \theta \frac{1}{M \cdot t} \sum_{s=1}^{M} \sum_{j=1}^{t} \left( \tilde{h}_j^{(s)} - h_j \right)^2$$
$$= \frac{1}{N \cdot c} \sum_{r=1}^{N} \sum_{i=1}^{c} \left( g_0 \left( \mathbf{a}_i^{(r)} \right) - y_i \right)^2$$
$$+ \theta \frac{1}{M \cdot t} \sum_{s=1}^{M} \sum_{j=1}^{t} \left( g_1 \left( \mathbf{b}_j^{(s)} \right) - h_j \right)^2. \tag{18}$$

Here, $\theta$ is the coefficient that controls how the treatment networks impact on the training process. In particular, if $\theta = 0$, then only the control network is trained on the controls without the treatments.

In sum, we achieve our first goal to train subnetworks implementing the kernels in the Nadaraya–Watson regression by using examples from the control and treatment groups. The trained kernels take into account the structures of the treatment and control data. The next task is to estimate CATE by using the trained kernels for some new vectors $\mathbf{x}$ or $\mathbf{z}$. It should be noted that all subnetworks can be represented as a single network due to the shared weights. In this case, arbitrary batches of pairs $(\mathbf{x}, \mathbf{x}_i)$ and pairs $(\mathbf{z}, \mathbf{z}_j)$ can be fed to the single network. This implies that we can construct testing neural networks consisting of $c$ and $t$ trained subnetworks in order to estimate $\tilde{y}$ and $\tilde{h}$ corresponding to $\mathbf{z}$ and $\mathbf{x}$, respectively, under condition $\mathbf{z} = \mathbf{x}$. Figures 4 and 5 show the trained neural networks for estimating $\tilde{y}$ and $\tilde{h}$, respectively. It is important to point out that the testing networks are not trained. Sets of $c$ pairs $(\mathbf{x}, \mathbf{x}_1), \ldots, (\mathbf{x}, \mathbf{x}_c)$ and $t$ pairs $(\mathbf{z}, \mathbf{z}_1), \ldots, (\mathbf{z}, \mathbf{z}_t)$ are fed to the subnetworks of the control and treatment networks, respectively. The whole examples for testing taking into account the outcomes are

$$\mathbf{a}(\mathbf{x}) = (\mathbf{x}_1, \ldots, \mathbf{x}_c, \mathbf{x}, y_1, \ldots, y_c), \tag{19}$$

and

$$\mathbf{b}(\mathbf{z}) = (\mathbf{z}_1, \ldots, \mathbf{z}_c, \mathbf{z}, h_1, \ldots, h_t). \tag{20}$$

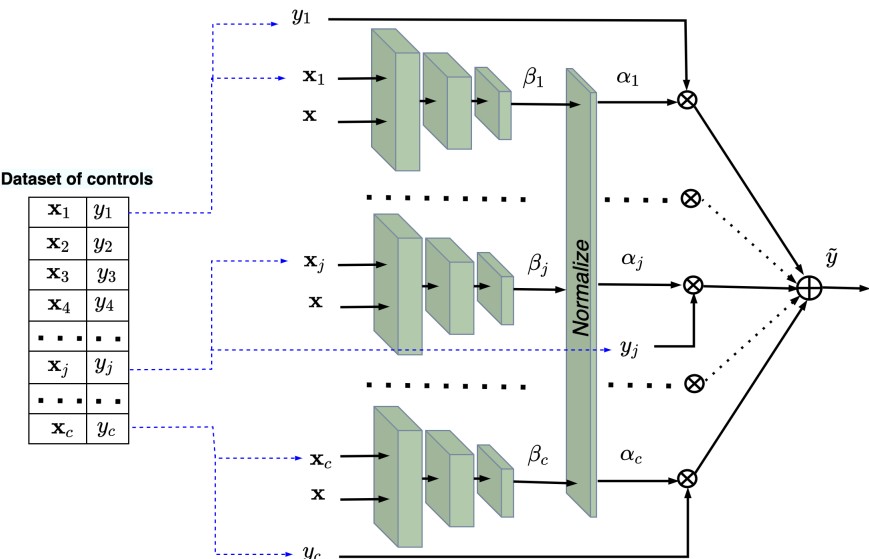

**Figure 4.** The trained neural network for computing $\tilde{y}$ in accordance with the Nadaraya–Watson regression.

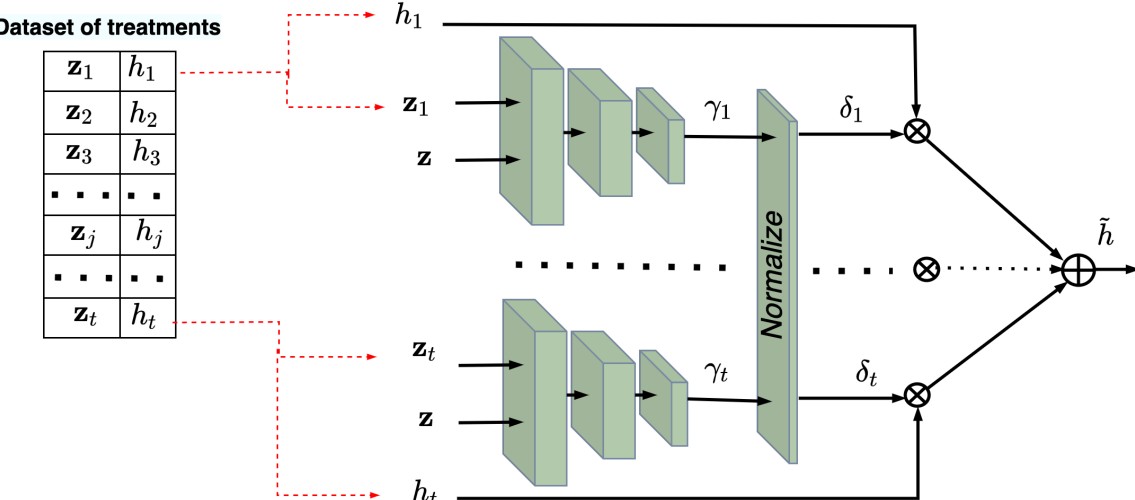

**Figure 5.** The trained neural network for computing $\tilde{h}$ in accordance with the Nadaraya–Watson regression.

In sum, the networks implement the Nadaraya–Watson regressions:

$$\tilde{y} = g_0(\mathbf{x}) = \sum_{j=1}^{c} \alpha(\mathbf{x}, \mathbf{x}_j) y_j,$$

$$\tilde{h} = g_1(\mathbf{z}) = \sum_{i=1}^{t} \delta(\mathbf{z}, \mathbf{z}_i) h_i. \tag{21}$$

Finally, CATE $\tau(\mathbf{x})$ or $\tau(\mathbf{z})$ is estimated as $\tau(\mathbf{x}) = \tau(\mathbf{z}) = \tilde{h} - \tilde{y}$.

The phases of the neural network training and testing are schematically shown as Algorithms 1 and 2, respectively.

---

**Algorithm 1** The algorithm implementing TNW-CATE in the training phase.

---

**Require:** Datasets $\mathcal{C}$ of $c$ controls and $\mathcal{T}$ of $t$ treatments, numbers $N$ and $M$ of generated subsets of $\mathcal{C}$ and $\mathcal{T}$, numbers of examples in generated subsets $n$ and $m$
**Ensure:** The trained weight sharing neural network implementing the Nadaraya–Watson regressions for control and treatment data
1: **for** $i = 1, i \le c$ **do**
2:      **for** $r = 1, r \le N$ **do**
3:          Generate subset $\mathcal{C}_{i,r} \subset \mathcal{C} \backslash (\mathbf{x}_i, y_i)$ (see (10))
4:          Form example $\mathbf{a}_i^{(r)}$ (see (11))
5:      **end for**
6: **end for**
7: **for** $j = 1, j \le t$ **do**
8:      **for** $s = 1, s \le M$ **do**
9:          Generate subset $\mathcal{T}_{j,s} \subset \mathcal{T} \backslash (\mathbf{z}_j, h_j)$ (see (12))
10:        Form example $\mathbf{b}_j^{(s)}$ (see (13))
11:      **end for**
12: **end for**
13: Train the weight sharing neural network with the loss function given in (18) on the set of pairs $(\mathbf{a}_i^{(r)}, \mathbf{b}_j^{(s)})$

---

---

**Algorithm 2** The algorithm implementing TNW-CATE in the testing phase.

---

**Require:** Trained neural control and treatment networks implementing the Nadaraya–Watson regressions; datasets $\mathcal{C}$ and $\mathcal{T}$; testing example $\mathbf{x} = \mathbf{z}$
**Ensure:** CATE $\tau(\mathbf{x})$
  1: Form the testing example $\mathbf{a}(\mathbf{x})$ in accordance with (19)
  2: Form the testing example $\mathbf{b}(\mathbf{z})$ in accordance with (20)
  3: Feed $\mathbf{a}(\mathbf{x})$ to the control network and $\mathbf{b}(\mathbf{z})$ to the treatment network and obtain the corresponding predictions $\tilde{y}$ and $\tilde{h}$
  4: $\tau(\mathbf{x}) = \tilde{h} - \tilde{y}$

---

It is important to point out that the neural networks shown in Figures 4 and 5 are not trained on datasets $\mathcal{C}$ and $\mathcal{T}$. These datasets are used as testing examples. This is an important difference between the proposed approach and other classification or regression models.

Let us return to the case $\theta = 0$ when only the control network is trained on the controls without the treatments. This case is interesting because it clearly demonstrates the transfer learning model when domains of source and target data are the same, but the tasks are different. Indeed, we train the kernels of the Nadaraya–Watson regression on the controls under the assumption that the domains of the controls and treatments are the same. Kernels learn the feature vector location. Actually, kernels are trained on controls by using outcomes $y_i$. However, nothing prevents us from using the same kernels with different outcomes $h_i$ if the structures of the feature vectors in the control and treatment groups are similar. We often use the same standard kernel with the same parameters, for example, the Gaussian one with the temperature parameter in machine learning tasks. The proposed method does the same, but with a more complex kernel.

## 5. Numerical Experiments

In this section, we provide simulation experiments evaluating the performance of meta-models for CATE estimation. In particular, we compare the T-learner, the S-learner, the X-learner, and the proposed method in several simulation studies. Numerical experiments are based on a random generation of the control and treatment data in accordance with different predefined controls and treatment outcome functions because the true CATEs are unknown due to the fundamental problem of the causal inference for real data [13].

### 5.1. General Parameters of Experiments

5.1.1. CATE Estimators for Comparison

The following models are used for their comparison with TNW-CATE.

1.  The T-learner [25] is a simple procedure based on estimating the control $g_0(\mathbf{x})$ and treatment $g_1(\mathbf{z})$ outcome functions by applying a regression algorithm, for example, the random forest [96]. The CATE in this case is defined as the difference $g_1(\mathbf{z}) - g_0(\mathbf{x})$.
2.  The S-learner was proposed in [25] to overcome difficulties and disadvantages of the T-learner. The treatment assignment indicator $T_i$ in the S-learner is included as an additional feature to the feature vector $\mathbf{x}$. The corresponding training set in this case is modified as $\mathcal{D}^* = \{(\mathbf{x}_1^*, y_1), \ldots, (\mathbf{x}_{c+t}^*, y_{c+t})\}$, where $\mathbf{x}_i^* = (\mathbf{x}_i, T_i) \in \mathbb{R}^{d+1}$ if $T_i = 0$, $i = 1, \ldots, c$, and $\mathbf{x}_{c+i}^* = (\mathbf{z}_i, T_i) \in \mathbb{R}^{d+1}$ if $T_i = 1$, $i = 1, \ldots, t$. Then the outcome function $g(\mathbf{x}, T)$ is estimated by using the training set $\mathcal{D}^*$. The CATE is determined in this case as $\tau(\mathbf{x}) = g(\mathbf{x}, 1) - g(\mathbf{x}, 0)$.
3.  The X-learner [25] is based on computing the so-called imputed treatment effects and is represented in the following three steps. First, the outcome functions $g_0(\mathbf{x})$ and $g_1(\mathbf{z})$

are estimated using a regression algorithm, for example, the random forest. Second, the imputed treatment effects are computed as follows:

$$D_1(\mathbf{z}_i) = h_i - g_0(\mathbf{z}_i),$$
$$D_0(\mathbf{x}_i) = g_1(\mathbf{x}_i) - y_i. \tag{22}$$

Third, two regression functions $\tau_1(\mathbf{z})$ and $\tau_0(\mathbf{x})$ are estimated for imputed treatment effects $D_1(\mathbf{z}_i)$ and $D_0(\mathbf{x}_i)$, respectively. CATE for a point $\mathbf{x} = \mathbf{z}$ is defined as a weighted linear combination of the functions $\tau_1(\mathbf{z})$ and $\tau_0(\mathbf{x})$ as $\tau(\mathbf{x}) = \varkappa \tau_0(\mathbf{x}) + (1 - \varkappa)\tau_1(\mathbf{x})$, where $\varkappa \in [0,1]$ is a weight which is equal to the ratio of treated patients [13].

### 5.1.2. Base Models for Implementing Estimators

Two models are used as the base regressors $g_0(\mathbf{x})$ and $g_1(\mathbf{z})$, which realize different CATE estimators for comparison purposes.

1. The first one is the random forest. It is used as the base regressor to implement the other models for two main reasons. First, we consider the case of the small number of treatments, and usage of neural networks does not allow us to obtain the desirable accuracy of the corresponding regressors. Second, we deal with tabular data for which it is difficult to train a neural network and the random forest is preferable. Parameters of the random forests used in the experiments are as follows:

   - Numbers of trees are 10, 50, 100, and 300;
   - Depths are 2, 3, 4, 5, 6, and 7;
   - The smallest values of examples which fall in a leaf are 1 example, 5%, 10%, and 20% of the training set.

   The above values for the hyperparameters are tested, choosing those leading to the best results.

2. The second base model used for realization different CATE estimators is the Nadaraya–Watson regression with the standard Gaussian kernel. This model is used because it is interesting to compare it with the proposed model that is also based on the Nadaraya–Watson regression but with the trainable kernel in the form of the neural network of the special form. Values $10^i$, $i = -8, \dots, 10$, and values 0.5, 5, 50, 100, 200, 500, and 700 of the bandwidth parameter $\gamma$ are tested, choosing those leading to the best results.

We use the following notation for different models depending on the base models and learners:

- **T-RF, S-RF, and X-RF** are the T-learner, the S-learner, and the X-learner with random forests as the base regression models;
- **T-NW, S-NW, and X-NW** are the T-learner, the S-learner, and the X-learner with the Nadaraya–Watson regression using the standard Gaussian kernel as the base regression models.

### 5.1.3. Other Parameters of Numerical Experiments

The mean squared error (MSE) as a measure of the regression model accuracy is used. For estimating MSE, we perform several iterations of the experiments such that each iteration is defined by the randomly selected parameters of experiments. MSE is computed by using 1000 points. In all the experiments, the number of treatments is 10% of the number of controls. For example, if 100 controls are generated for an experiment, then 10 treatments are generated in addition to controls such that the total number of examples is 110. After generating the training examples, their responses $y$ are normalized, but the corresponding initial mean and the standard deviation of responses are used to normalize responses of the test examples. This procedure allows us to reduce the variance among results at different iterations. The generated feature vectors in all experiments consist of 10 features. To select optimal hyperparameters of all regressors, additional validation

examples are generated such that the number of controls is 20% of the training examples from the control group.

### 5.1.4. Functions for Generating Datasets

The following functions are used to generate synthetic datasets:

1.  **Spiral functions:** The functions are named spiral because when using two features vectors they are located on the Archimedean spiral. For even $d$, we write the vector of features as

$$\mathbf{x} = (t\sin(t), t\cos(t), \dots,$$
$$t\sin(t \cdot d/2), t\cos(t \cdot d/2)). \tag{23}$$

For odd $d$, there holds

$$\mathbf{x} = (t\sin(t), t\cos(t), \dots, t\sin(t \cdot \lceil d/2 \rceil)). \tag{24}$$

The responses are generated as a linear function of $t$, i.e., they are computed as $y = at + b$.

Values of parameters $a$, $b$, and $t$ for performing numerical experiments with spiral functions areas follows:

- The control group: parameters $a$, $b$, and $t$ are uniformly generated from intervals $[1, 4]$, $[1, 4]$, and $[0, 10]$, respectively.
- The treatment group: parameters $a$, $b$, and $t$ are uniformly generated from intervals $[8, 10]$, $[8, 10]$, and $[0, 10]$, respectively.

2.  **Logarithmic functions:** Features are logarithms of the parameter $t$, i.e., there holds

$$\mathbf{x} = (a_1 \ln(t), a_2 \ln(t), \dots, a_d \ln(t)). \tag{25}$$

The responses are generated as a logarithmic function with adding an oscillating term $b\sin(t)$ to $y$, i.e., there holds $y = b(\ln(t) + \sin(t))$.

Values of parameters $a_1, \dots, a_d$, $b$ for performing numerical experiments with logarithmic functions are as follows:

- Each parameter from the set $\{a_1, \dots, a_d\}$ is uniformly generated from intervals $[-4, -1] \cup [1, 4]$ for controls as well as for treatments.
- Parameter $b$ is uniformly generated from interval $[1, 4]$ for controls and from interval $[-4, -1]$ for treatments.
- Values of $t$ are uniformly generated in interval $[0.5, 5]$.

3.  **Power functions:** Features are represented as powers of $t$. For arbitrary $d$, the vector of features is represented as

$$\mathbf{x} = (t^{1/\sqrt{d}}, t^{2/\sqrt{d}}, \dots, t^{d/\sqrt{d}}). \tag{26}$$

However, features which are close to linear ones, e.g., $x_i = t^{i/\sqrt{d}}$ for $0.8 < i/\sqrt{d} < 1.6$, are replaced with the Gaussian noise having the unit standard deviation and the zero expectation, i.e., $x_i \sim \mathcal{N}(0, 1)$. The responses are generated as follows:

$$y = a \cdot \exp\left(-\frac{(t - s)^2}{b}\right). \tag{27}$$

Values of parameters $a$, $b$, $s$, and $t$ for performing numerical experiments with power functions are as follows:

- The control group: parameters $a$ and $b$ are uniformly generated from intervals $[1, 2]$ and $[0.25, 1]$, respectively; parameter $s$ is 2.5.

- The treatment group: parameters $a$ and $b$ are uniformly generated from intervals $[2, 4]$ and $[1, 2]$, respectively; parameter $s$ is 2.5.
- Values of $t$ are uniformly generated in interval $[0, 5]$.

4. **Indicator functions** [25]: The functions are expressed through the indicator function $I$ taking value 1 if its argument is true.

- The function for controls is represented as

$$g_0(\mathbf{x}) = \mathbf{x}^{\mathrm{T}}\beta + 5I(x_1 > 0.5). \tag{28}$$

- The function for treatments is represented as

$$g_1(\mathbf{x}) = \mathbf{x}^{\mathrm{T}}\beta + 5I(x_1 > 0.5) \\ +8I(x_2 > 0.1). \tag{29}$$

- Vector $\beta$ is uniformly distributed in interval $[-5; 5]^d$; values of features $x_i = 1, \ldots, d$, are uniformly generated from interval $[-1, 1]$.

The indicator function differs from other functions considered in numerical examples. It is taken from [25] in order to study TNW-CATE when the assumption of specific and similar domains for the control and treatment feature vectors can be violated.

In numerical experiments with the above functions, we take parameter $\theta$ to be equal to 0.1, 0.5, 0.5, and 0.5 for the spiral, logarithmic, power, and indicator functions, respectively, except for experiments which study how parameter $\theta$ impacts the MSE.

### 5.2. Study of the TNW-CATE Properties

In all figures illustrating dependencies of CATE estimators on parameters of models, dotted curves correspond to the T-learner, the S-learner, and the X-learner implemented by using the Nadaraya–Watson regression (triangle markers correspond to T-NW and S-NW; the circle marker corresponds to X-NW). Dashed curves with the same markers correspond to the same models implemented by using random forests. The solid curve with cross markers corresponds to TNW-CATE.

#### 5.2.1. Experiments with Numbers of Training Data

Let us compare different CATE estimators using different numbers of the control and treatment examples. We study the estimators by numbers $c$ of controls: 100, 250, 500, 750, and 1000. Each number of treatments is determined as 10% of each number of controls. Value $n$ is 80 and 100; value $m$ is 50% of $t$. Figure 6 illustrates how MSE of the CATE values depends on the number $c$ of controls for different estimators when different functions are used for generating examples. In fact, these experiments study how MSE depends on the entire number of the controls and treatments because the number of the treatments increases with the number of the controls. For all functions, the increase in the amount of training examples improves most estimators including TNW-CATE. These results are expected because the larger size of training data mainly leads to the better accuracy of models. It can be seen from Figure 6 that the proposed model provides better results in comparison with other models. The best results are achieved when the spiral generating function is used. The models different from TNW-CATE cannot cope with the complex structure of data in this case. However, TNW-CATE shows comparative results with the T-learner, the S-learner, and the X-learner when the indicator function is used for generating examples. The X-learner outperforms TNW-CATE in this case. The indicator function does not have a complex structure. Moreover, the corresponding outcomes linearly depend on most features (see (28) and (29)), and random forests implementing X-RF are trained better than the neural network implementing TNW-CATE. One can also see from Figure 6 that models T-NW, S-NW, and X-NW based on the Nadaraya–Watson regression with the Gaussian kernel provide close results when the logarithmic generating function is used by larger numbers of training data. This is caused

by the fact that the Gaussian kernel is close to the neural network kernel implemented in TNW-CATE.

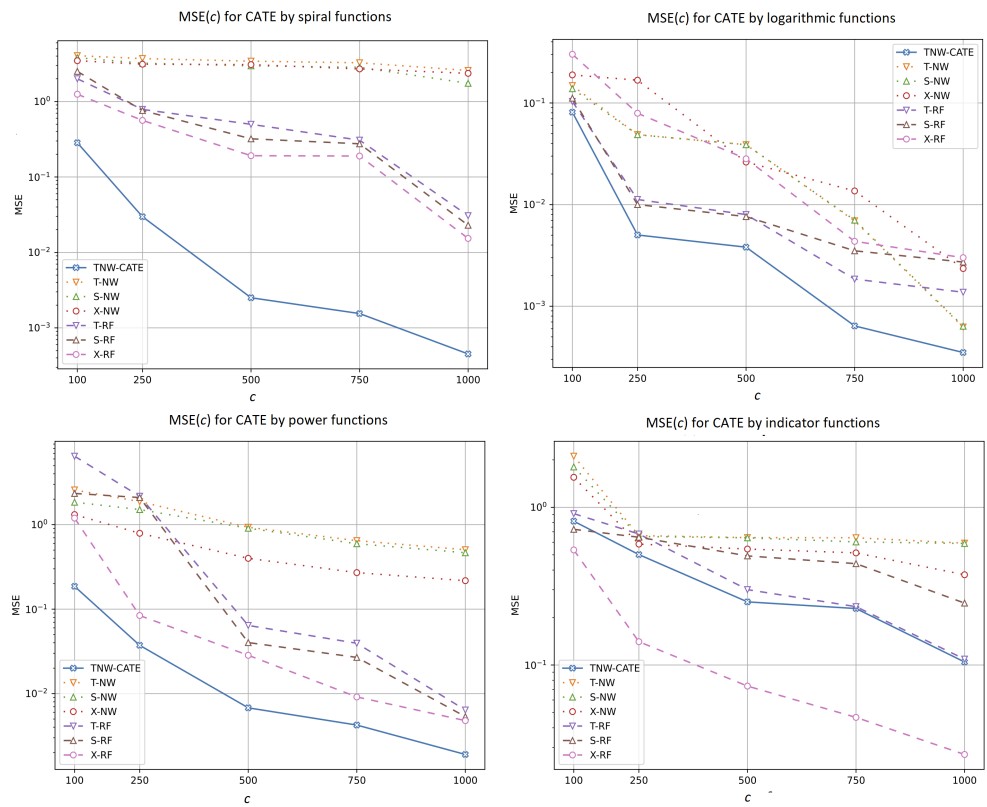

**Figure 6.** MSE of the CATE values as a function of the number of controls when spiral, logarithmic, power, and indicator functions are used for generating examples.

Figure 7 illustrates how the number of controls separately impacts the control (the left plot) and treatment (the right plot) regressions when the power function is used for generating data. It can also be seen from Figure 7 that MSE provided by the control network is much smaller than MSE of the treatment neural network.

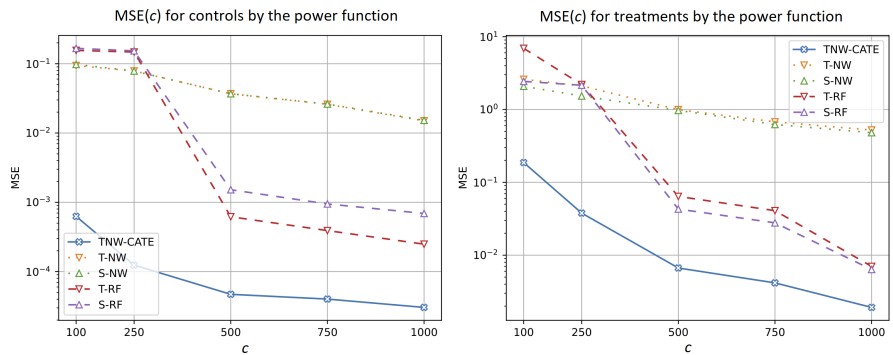

**Figure 7.** MSE of the control (the **left** plot) and treatment (the **right** plot) responses as functions of the number of controls when the power function is used for generating examples.

### 5.2.2. Experiments with Different Values of the Treatment Ratio

Another question is how the CATE estimators depend on the ratio of numbers of treatments and controls in the training set. We study the case when the number of controls $c$ is 200 and the ratio takes values from the set $\{0.1, 0.2, 0.3, 0.4, 0.5\}$. The coefficient $\theta$ is taken in accordance with the certain function as described above.

Similar results are shown in Figure 8 where plots of MSE of the CATE values as a function of the ratio of numbers of treatments in the training set by different generating functions are depicted. We again see from Figure 8 that the difference between MSE of TNW-CATE and other models is the largest when the spiral function is used. TNW-CATE also provides better results in comparison with other models, except for the case of the indicator function when is TNW-CATE inferior to the X-RF.

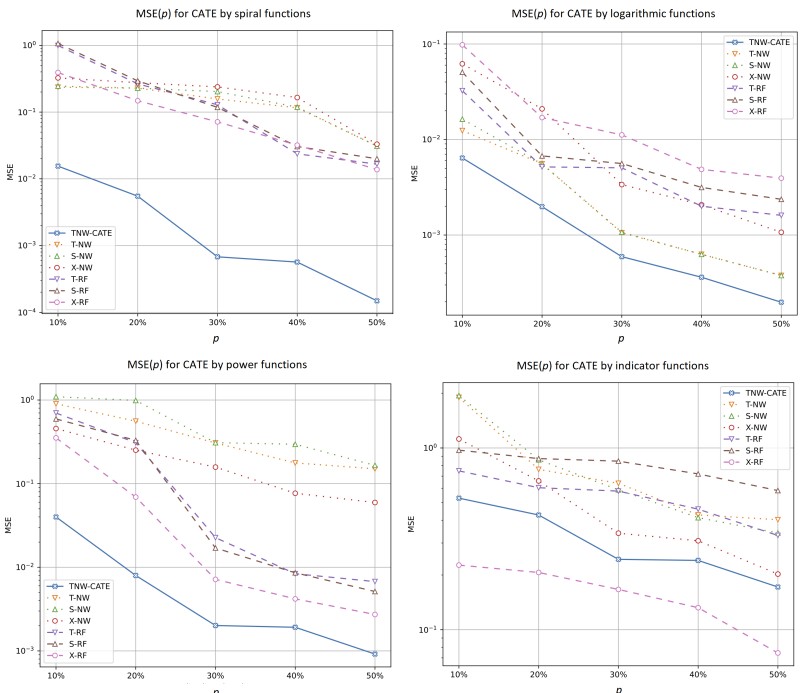

**Figure 8.** MSE of the CATE values as a function of the ratio of numbers of treatments in the training set when spiral, logarithmic, power, and indicator functions are used for generating examples.

Figure 9 illustrates how the ratio of numbers of the treatments separately impacts the control (the left plot) and treatment (the right plot) regressions when the logarithmic function is used for generating data. It can be seen from Figure 9 that MSE of the treatment network is very close to MSE of other models for almost all values of the ratio, but the accuracy of the control network significantly differs from other models.

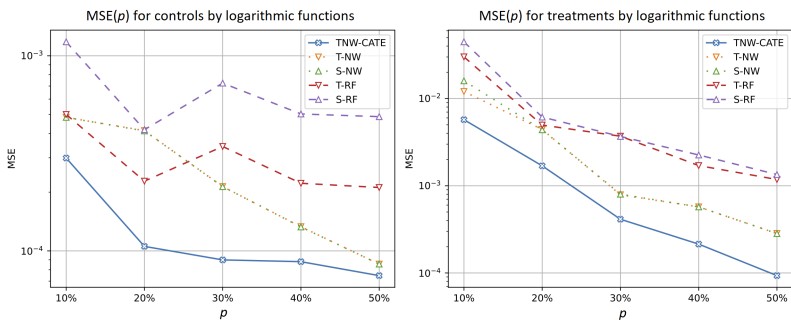

**Figure 9.** MSE of the control (the **left** plot) and treatment (the **right** plot) responses as functions of the ratio of numbers of treatments in the training set when the logarithmic function is used for generating examples.

5.2.3. Experiments with Different Values of $\theta$

The next experiments allow us to investigate how the CATE estimators depend on the value of hyperparameter $\theta$ which controls the impact of the control and treatment

networks in the loss function (18). The corresponding numerical results are shown in Figures 10 and 11. It should be noted that other models do not depend on $\theta$. One can see from Figures 10 and 11 that there is an optimal value of $\theta$ minimizing MSE of TNW-CATE for every generating function. For example, the optimal $\theta$ for the spiral function is 0.1. It can be seen from Figure 10 that TNW-CATE can be inferior to other models when $\theta$ is not optimal. For example, the case $\theta = 0$ for the logarithmic function leads to worse results for TNW-CATE in comparison with T-RF and S-RF.

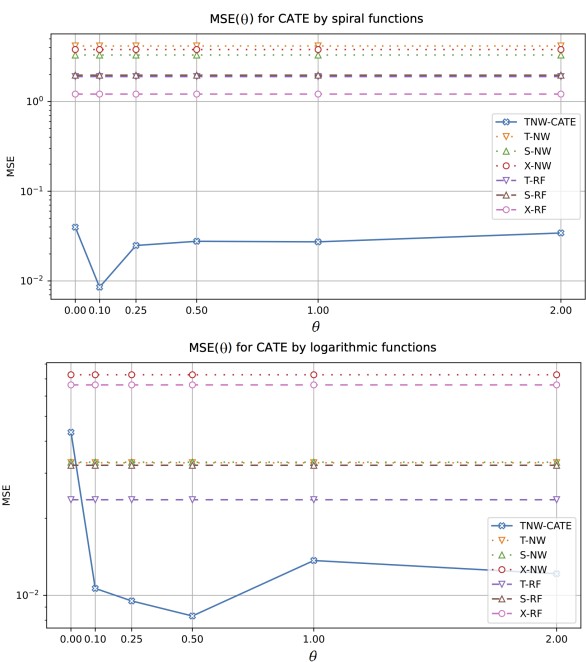

**Figure 10.** MSE of the CATE values as a function of the coefficient $\theta$ when spiral (the **first** plot) and logarithmic (the **second** plot) functions are used for generating examples.

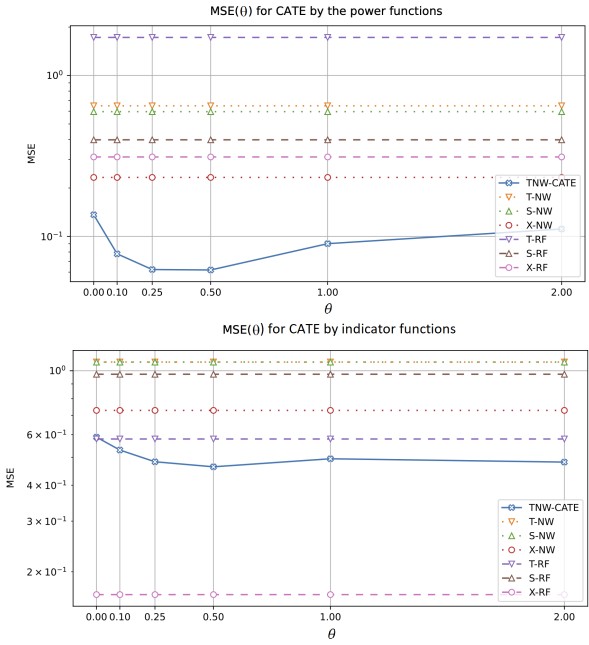

**Figure 11.** MSE of the CATE values as a function of the coefficient $\theta$ when power (the **first** plot) and indicator (the **second** plot) functions are used for generating examples.

Numerical results are also presented in Table 1 where the MSE values and standard deviations corresponding to different models using different generating functions are given. The best results for every function are shown in bold. It can be seen from Table 1 that TNW-CATE provides the best results for the spiral, logarithmic, and power functions. Moreover, the improvement is sufficient. TNW-CATE is comparable with the X-NW and X-RF from the logarithmic and power generating functions. At the same time, the proposed model is inferior to X-RF for the indicator function.

**Table 1.** The best MSE values and standard deviations of CATE for different models and by different generating functions.

| | Functions | | | |
|---|---|---|---|---|
| **Model** | **Spiral** | **Logarithmic** | **Power** | **Indicator** |
| T-NW | $3.806 \pm 0.734$ | $0.377 \pm 0.163$ | $1.722 \pm 0.482$ | $0.650 \pm 0.208$ |
| S-NW | $3.629 \pm 0.605$ | $0.341 \pm 0.057$ | $1.719 \pm 0.287$ | $0.549 \pm 0.092$ |
| X-NW | $3.279 \pm 0.547$ | $0.542 \pm 0.090$ | $0.632 \pm 0.105$ | $0.353 \pm 0.059$ |
| T-RF | $2.278 \pm 0.380$ | $0.051 \pm 0.019$ | $2.743 \pm 0.457$ | $0.337 \pm 0.056$ |
| S-RF | $2.575 \pm 0.409$ | $0.060 \pm 0.018$ | $0.839 \pm 0.140$ | $0.434 \pm 0.072$ |
| X-RF | $1.385 \pm 0.231$ | $0.202 \pm 0.074$ | $0.805 \pm 0.134$ | $\mathbf{0.061} \pm 0.010$ |
| TNW-CATE | $\mathbf{0.232} \pm 0.069$ | $\mathbf{0.026} \pm 0.008$ | $\mathbf{0.353} \pm 0.089$ | $0.257 \pm 0.043$ |

We can find many successful applications of neural networks for medical and biological tasks [97,98]. Neural networks also are used in the CATE estimation, for example, DRNet [72], DragonNet [14], FlexTENet [66], and VCNet [69]. However, it is important to point out that many models based on neural networks have not been successful when they are trained on small datasets because the aforementioned neural networks require a large amount of data for training. Thus, the considered small datasets have led to the network overfitting. Therefore, we have studied models for comparison that are based on methods dealing with small data.

*5.3. Real Dataset*

We perform numerical experiments on the popularly used datasets from the "The Infant Health and Development Program"(IHDP), which are designed to understand the effect of home visits by specialist doctors on the future cognitive test scores of premature infants [8]. The dataset can be regarded as a common benchmarking dataset for estimating HTE. It contains 747 subjects and 25 features (6 continuous and 19 binary features) that describe both the characteristics of the infants and the characteristics of their mothers. The datasets can be accessed at https://github.com/vdorie/npci (accessed on 19 April 2023). The number of treatments is 139. Results of numerical experiments (the MSE values and standard deviations) with IHDP by using the same models as in the previous experiments are presented in Table 2. We also show the MSE values and standard deviations of different numbers of treatments (70 and 35) by randomly removing a part of treatments from the dataset. We randomly split the dataset into three subsets: training, validation, and testing. We use 40% of the examples for testing. The threefold validation based on 20% of examples is used to tune the model parameters during training. It can be seen from Table 2 that TNW-CATE provides the smallest values of the MSE. It is important to note that the models with the Gaussian kernel (T-NW, S-NW, and X-NW) provide the worst results. This example illustrates a case when the data structure is too complex to model by means of the standard Gaussian kernel.

**Table 2.** The best MSE values and standard deviations of CATE for different models by studying the dataset IHDP.

| | Number of Treatments | | |
|---|---|---|---|
| **Model** | **139** | **70** | **35** |
| T-NW | $0.482 \pm 0.172$ | $0.511 \pm 0.138$ | $0.696 \pm 0.209$ |
| S-NW | $0.277 \pm 0.138$ | $0.177 \pm 0.079$ | $0.304 \pm 0.108$ |
| X-NW | $0.276 \pm 0.014$ | $0.240 \pm 0.096$ | $0.348 \pm 0.139$ |
| T-RF | $0.099 \pm 0.039$ | $0.218 \pm 0.082$ | $0.231 \pm 0.057$ |
| S-RF | $0.062 \pm 0.028$ | $0.067 \pm 0.015$ | $0.075 \pm 0.037$ |
| X-RF | $0.046 \pm 0.021$ | $0.063 \pm 0.014$ | $0.079 \pm 0.034$ |
| TNW-CATE | $\mathbf{0.010} \pm 0.006$ | $\mathbf{0.021} \pm 0.007$ | $\mathbf{0.038} \pm 0.019$ |

After analyzing the obtained numerical results, we can summarize the following:

1. The results significantly depend on the structure of the control and treatment data. The most complex structure among the considered ones is produced by the spiral function, and we observe that TNW-CATE outperforms the results in comparison with other models, especially when the number of controls is rather large. The large number of controls prevents the neural network from overfitting. The worst results provided by TNW-CATE and models based on the Gaussian kernels (T-NW, S-NW, and X-NW) for the indicator function are caused by the fact that the random forest can be regarded as one of the best models for the functions such as the indicator. The neural network cannot cope with this structure due to the properties of its training.

2. It can be seen from the experiments that the difference between TNW-CATE and other models increases as the number of treatments decreases. It does not mean that the MSE of TNW-CATE is increased. The MSE is decreased. However, this decrease is not as significant as in other models. This is caused by joint use of controls and treatments in training the neural network. It follows from the above that the co-training of the neural network on treatments corrects the network weights when the domain of the treatment group is shifted relative to the control domain.

3. The main conclusion from the experiments using the dataset IHDP is that the neural network is perfectly adapted to the data structure which is totally unknown. In other words, the neural network tries to implement is own distance function and a form of the kernel to fit the data structure. This is an important distinction of TNW-CATE from other approaches.

## 6. Conclusions

A new method (TNW-CATE) for solving the CATE problem has been studied. The main idea behind TNW-CATE is to use the Nadaraya–Watson regression with kernels that are implemented as neural networks of the considered specific form and are trained on many randomly selected subsets of the treatment and control data. With the proposed method, we aimed to avoid constructing the regression function $g_1$ based only on the treatment group because it may be small. Moreover, we aimed to avoid using standard kernels, for example, Gaussian ones, in the Nadaraya–Watson regression. By training kernels on controls (or controls and treatments), we aimed to transfer knowledge of the feature vector structure in the control group to the treatment group.

The proposed model and the learning algorithm can be used in many applications, for example, for drug development and related steps such as clinical trials and study design or selection of the optimal dose of medicine for a patient, etc. This model, along with other models, can be regarded as a basis for personalized medicine where the need for personalized treatment is tremendous. Taking into account all the characteristics of patients can make the treatment of many diseases more successful. When new treatments or new drugs are tested, the number of treatments may be very small. In this case, the proposed model can be effective. Another application of the model is a case when the feature vectors

characterizing patients have a complex structure which cannot be appropriately represented by typical regression functions or kernels. One of the sources of the complex structure is the multimodal information about patients, for example, computer tomography, electronic health records, etc.

The main limitation of the proposed model is the need to tune the hyperparameters of the neural network, which requires significant time spent on network training and validation. There are no strict heuristics that can significantly speed up this stage. However, this time is not crucial. For instance, the time of training TNW-CATE on the dataset IHDP (see Section 5.3) is 1704 s, whereas the times of training the models T-RF, S-RF, and X-RF are 49 s, 68 s, and 114 s, respectively. With this time complexity, we pay for the accuracy of the results. In spite of the apparent complexity of the whole neural network for training, TNW-CATE is actually simple because it can be realized as a single small subnetwork implementing the kernel.

The numerical simulation experiments have illustrated the outperformance of TNW-CATE for several datasets.

We used neural networks to learn kernels in the Nadaraya–Watson regression. However, different models can be applied to the kernel implementation, for example, the random forest [96] or the gradient boosting machine [99]. The study of different models for estimating CATE is a potential direction for further research. We also assumed the similarity between structures of control and treatment data. This assumption can be violated in some cases. Therefore, it would be interesting to modify TNW-CATE to account for these violations. This is another potential direction for further research. Another interesting direction for potential future research is to incorporate robust procedures and imprecise statistical models to deal with small datasets into TNW-CATE. The incorporation of the models can provide estimates for CATE, which are more robust than estimates obtained by using TNW-CATE.

**Author Contributions:** Conceptualization, L.U. and A.K.; methodology, L.U. and A.K.; software, S.K.; validation, S.K. and A.K.; formal analysis, L.U. and A.K; investigation, A.K. and S.K.; resources, L.U. and S.K.; data curation, A.K. and S.K.; writing—original draft preparation, L.U. and S.K.; writing—review and editing, A.K.; visualization, S.K.; supervision, L.U.; project administration, L.U.; funding acquisition, L.U. All authors have read and agreed to the published version of the manuscript.

**Funding:** This work is supported by the Russian Science Foundation under grant 21-11-00116.

**Institutional Review Board Statement:** Not applicable.

**Informed Consent Statement:** Not applicable.

**Data Availability Statement:** Not applicable.

**Acknowledgments:** The authors would like to express their appreciation to the anonymous referees whose very valuable comments have improved the paper.

**Conflicts of Interest:** The authors declare no conflict of interest.

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
