# Peer review of "Heterogeneous Treatment Effect with Trained Kernels of the Nadaraya–Watson Regression"

_algorithms, doi:10.3390/a16050226_

Round 1

Reviewer 1 Report

A new method to solve the CATE problem based on neural networks to implement the NW regression.

The paper is well written, the motivation/introduction is clear with a very long list of references and the numerical experiments include different signals/algorithms and evaluation of some hyperparameters.

There would be an addition that will make the paper not only interesting from a theoretical/algorithmic  point of view, but useful in real world: It would be nice if authors can add some results for real data, not only numerical simulations. 

Author Response

Comment: There would be an addition that will make the paper not only interesting from a theoretical/algorithmic  point of view, but useful in real world: It would be nice if authors can add some results for real data, not only numerical simulations. 

Response: Numerical experiments are based on random generation of control and treatment data in accordance with different predefined control and treatment outcome functions because the true CATEs are unknown due to the fundamental problem of the causal inference for real data. However, there is a benchmarking dataset "The Infant Health and Development Program" (IHDP) which is designed to understand the effect of home visits by specialist doctors on the future cognitive test scores of premature infants. We have introduced numerical experiments with this dataset into Section 5 as an additional subsection called “Real dataset”. In this subsection, the comparison results are provided in Table 2.

Reviewer 2 Report

In the introduction, what key theoretical perspectives and empirical findings in the main literature have already informed the problem formulation? What major, unaddressed puzzle, controversy, or paradox does this research address? 

Why does it need to be addressed?

Why it should be now - not in the past?

Further, in the introduction, what is the recent knowledge gap of the main literature that the author needs to write this research? What we have known and what we have not known? What is missing from current works? Please explain and give examples!

In terms of the knowledge gap, it will be best if the research challenge/knowledge gap could be stated in one article or more articles in the main literature (optional). Assure that you have included all key articles (e.g., most widely cited articles) in the main literature. Mention them.

Theoretical contribution to the main literature seems to be weak. How does the research fulfil the knowledge gap discussed in the introduction? In terms of the knowledge gap, indicates what is the new, exciting, and not trivial contribution you offer.

• What are the practical implications of your research? 

Authors should further clarify and elaborate novelty in their contribution.

What are the limitations of the present work?

Author Response

Comment:  In the introduction, what key theoretical perspectives and empirical findings in the main literature have already informed the problem formulation? What major, unaddressed puzzle, controversy, or paradox does this research address? Why does it need to be addressed?

Response: We have added the following text to the Introduction: “…The first one is that the control group is usually larger than the treatment group. Hence, we meet the problem of a small training dataset, which does not allow us to apply many efficient machine learning methods directly. Aoki and Ester [11] consider an example of difficulties to collect the corresponding information about patients and to explore the side effects of drugs adopted on certain pediatric cancer treatments due to several reasons, including the sensitivity of issues for the families, the rarity of a disease, an effort between hospitals and doctors to refer the patients. The authors propose to use the transfer learning approach, implemented by means of a neural network, to estimate treatment effects on small datasets. This is an efficient and interesting approach. However, it may encounter difficulties when training the neural network which models the regression functions for controls and treatments. The same problem has been pointed out in [4,12]. Therefore, we need to develop a similar approach where the neural networks should be rather simple and play a secondary role in the modeling the regression functions.”

We also added: “… This is a fundamental problem of computing the causal effect. This problem is solved by the explicit or implicit construction of regression functions for the control and treatment patients. For instance, we can train neural networks which predict outcomes for a new patient under condition that this patient belongs to the control and treatment groups. However, this way requires to have large datasets to train the corresponding machine learning models. We return to the first problem of the treatment group size.”

We also added: “One of the difficulties is a complex data structure. A lot of methods use some assumptions about the model parameters and the data structure. For instance, a Gaussian mixture for outcomes is proposed to use in [11]. In some cases, these assumptions are correct, and they correspond to the real structure of the treatment and control data. However, they can lead to errors in other cases where the data structure is rather complex. Dorie et al. [16] note that the analysis of observations, having a grouped structure, shows that the impact of the treatment exposure will vary across these groups. In this case, most current machine learning approaches ignore these varying effects. Therefore, it is necessary to develop models that take into account these problems.”

We added the following references which state these problems:

  1. Xie, Y.; Brand, J.; Jann, B. Estimating Heterogeneous Treatment Effects with Observational Data. Sociological Methodology 2012, 42, 314–347.
  2. Caron, A.; Baio, G.; Manolopoulou, I. Estimating Individual Treatment Effects using Non-Parametric Regression Models: a Review. Journal of the Royal Statistical Society Series A: Statistics in Society 2022, 185, 1115–1149.
  3. Zhou, X.; Xie, Y. Heterogeneous Treatment Effects in the Presence of Self-Selection: A Propensity Score Perspective. Sociological Methodology 2020, 50, 350–385.
  4. Hill, J. Bayesian nonparametric modeling for causal inference. Journal of Computational and Graphical Statistics 2011, 20.
  5. Aoki, R.; Ester, M. Causal Inference from Small High-dimensional Datasets. arXiv:2205.09281.
  6. Alaa, A.; van der Schaar, M. Limits of Estimating Heterogeneous Treatment Effects: Guidelines for Practical Algorithm Design. In Proceedings of the Proceedings of the International Conference on Machine Learning. PMLR, 2018, pp. 129–138.
  7. Dorie, V.; Perrett, G.; Hill, J.; Goodrich, B. Stan and BART for Causal Inference: Estimating Heterogeneous Treatment Effects Using the Power of Stan and the Flexibility of Machine Learning. Entropy 2022, 24, 1782.

Comment: Further, in the introduction, what is the recent knowledge gap of the main literature that the author needs to write this research? What we have known and what we have not known? What is missing from current works? Please explain and give examples!

Response: Regression models of specific forms for controls and treatments are determined in most methods which are available in the literature. The models differ mainly by types of the regression functions and their parameters. In contrast to these models, the Nadaraya-Watson kernel regression model which does not rely on specific regression functions. It estimates regression values without any assumptions about the functions.

It should be noted that several authors [43-46] proposed to use the Nadaraya-Watson kernel regression with standard kernels having the bandwidth parameter to construct the CATE estimator. However, they do not propose to learn the kernels. Moreover, they do not propose to use the kernels in the framework of transfer learning where the kernels are trained on controls, but they are used for treatments. The Nadaraya-Watson kernel regression in these works has a relative disadvantage. It requires to define a certain kernel for computing weights of examples, for example, the Gaussian kernel. We overcome the above difficulty by replacing the standard kernel with a neural network which implements the kernel.

We have added the above to the Introduction section.

Comment: In terms of the knowledge gap, it will be best if the research challenge/knowledge gap could be stated in one article or more articles in the main literature (optional). Assure that you have included all key articles (e.g., most widely cited articles) in the main literature. Mention them.

Response: We have added the following references where the Nadaraya-Watson kernel regression is proposed to construct the CATE estimator:

  1. Gao, Z.; Han, Y. Minimax optimal nonparametric estimation of heterogeneous treatment effects. In Proceedings of the Advances in Neural Information Processing Systems, 2020, Vol. 33, pp. 21751–21762. 579
  2. Hsu, Y.C.; Lai, T.C.; Lieli, R. Counterfactual treatment effects: Estimation and inference. Journal of Business & Economic Statistics 2022, 40, 240–255. 581
  3. Padilla, O.; Yu, Y. Dynamic and heterogeneous treatment effects with abrupt changes. arXiv:2206.09092.
  4. Sun, X. Estimation of Heterogeneous Treatment Effects Using a Conditional Moment Based Approach. arXiv:2210.15829.

Comment: Theoretical contribution to the main literature seems to be weak. How does the research fulfil the knowledge gap discussed in the introduction? In terms of the knowledge gap, indicates what is the new, exciting, and not trivial contribution you offer. Authors should further clarify and elaborate novelty in their contribution.

Response: We have added the following list of contributions to the Introduction:

“Our contributions can be summarized as follows:

  1. We propose to use the Nadaraya-Watson kernel regression which does not rely on specific regression functions and estimates regression values (outputs of controls and treatments) without any assumptions about the functions. The main feature of the model is that kernels of the Nadaraya-Watson regression are implemented as neural networks, i.e. the kernels are trained on the control and treatment data. In contrast to many CATE estimators based on neural networks, the proposed model uses simple neural networks which implement only kernels, but not the regression functions.
  2. The proposed model and the neural network architecture allow us to solve the problem of small numbers of patients in the treatment group. This is a crucial problem especially when new treatments and new drugs are tested. In fact, the proposed model can be considered in the framework of the transfer learning when controls can be viewed as source data (in terms of the transfer learning), but treatments are target data.
  3. Neural networks implementing the kernels amplifies the model flexibility. In contrast to the standard kernels, the neural kernels allow us to cope with the possible complex data structure because they are adapted to the structure due to many trainable parameters.
  4. A specific algorithm of training the neural kernels is proposed. It trains networks on controls and treatments simultaneously in order to memorize the treatment data structure. We show by means of numerical examples that there is an optimal linear combination of two loss functions corresponding to the controls and treatments. “

Comment: What are the practical implications of your research? 

Response: We have added to the Conclusion section the following text: “The proposed model and the learning algorithm can be used in many applications, for example, to drug development, including clinical trials, study design, to selection of the optimal dose of medicine for a patient, etc. This model, along with other models, can be regarded as a basis for personalized medicine where the need of the personalized treatment is tremendous. Taking into account all characteristics of patients can make the treatment of many diseases successful. When new treatments or new drugs are tested, the number of treatments may be very small. In this case, the proposed model can be effective. Another application of the model is a case when the feature vectors characterizing patients have a complex structure which cannot be appropriately represented by typical regression functions or kernels. One of the sources of the complex structure is the multimodal information about patients, for example, the computer tomography, electronic health records, etc.”

Comment: What are the limitations of the present work?

We have added to the Conclusion section: “The main limitation of the proposed model is the need to tune the hyperparameters of the neural network, which requires significant time spent on network training and validation. There are no strict heuristics that can significantly speed up this stage.”

Reviewer 3 Report

In this study, the authors proposed a method for estimating the conditional average treatment effect. Although the idea is of interest, some major points should be addressed as follows:

1. Overall, English writing and presentation style should be improved to meet the publication standard.

2. The authors are suggested to conduct cross-validation during the training process.

3. More use cases should be added to validate.

4. Uncertainties of models should be reported.

5. Some results were listed without in-depth discussions.

6. Neural network is well-known and has been used in previous studies i.e., PMID: 30866734, PMID: 36166351. Thus, the authors are suggested to refer to more works in this description to attract a broader readership.

7. The authors mentioned "The code of proposed algorithms implementing TNW-CATE is publicly available.", but we could not see it.

Author Response

Comment: Overall, English writing and presentation style should be improved to meet the publication standard.

Response: We have tried to improve English writing.

Comment: The authors are suggested to conduct cross-validation during the training process.

Response: The cross-validation is used during training. It is written in Section 5: “To select optimal hyperparameters of all regressors, additional validation examples are generated such that the number of controls is 20% of the training examples from the control group.” It has been also written in Subsection “Real dataset”: “We split randomly the dataset into three subsets: training, validation, and testing. We use 40% of examples for testing. The 3-fold validation based on 20% of examples is used to tune the model parameters during training.”

Comment: More use cases should be added to validate.

Response: We have added numerical experiments with a benchmarking dataset "The Infant Health and Development Program" (IHDP) which is designed to understand the effect of home visits by specialist doctors on the future cognitive test scores of premature infants. We have introduced Numerical experiments with this dataset have been inserted in Section 5 as an additional subsection called “Real dataset”. In this subsection, the comparison results are provided in Table 2.

Comment: Uncertainties of models should be reported.

Response: We have added values of standard deviations to Table 1 and to Table 2 in order to take into account uncertainties of results provided by models.

Comment:  Some results were listed without in-depth discussions.

Response: We have added the following text to the end of Section 5:

“Analyzing the obtained numerical results, we can summarize the following:

  1. The results significantly depend on the structure of the control and treatment data. The most complex structure among the considered ones is produced by the spiral function, and we observe that TNW-CATE provides outperforming results in comparison with other models in this case, especially, when the number of controls is rather large. The large number of controls prevents the neural network from overfitting. The worse results provided by TNW-CATE as well as models based on the Gaussian kernels (T-NW, S-NW, X-NW) for the indicator function are caused by the fact that the random forest can be regarded as one of the best models for the functions like the indicator one. The neural network cannot cope with this structure due to properties of its training.
  2. It can be seen from the experiments that the difference between TNW-CATE and other models increases as the number of treatments decreased. It does not mean that the MSE of TNW-CATE is increased. The MSE is decreased. However, this decrease is not as significant as in other models. This is caused by joint use of controls and treatments in training the neural network. It follows from the above that the co-training of the neural network on treatments corrects the network weights when the domain of the treatment group is shifted relative to control domain.
  3. The main conclusion from the experiments with the dataset IHDP is that the neural network is perfectly adapted to the data structure which is totally unknown. In other words, the neural network tries to implement is own distance function and a form of the kernel to fit the data structure. This is an important distinction of TNW-CATE from other approaches.”

Response:

Comment: Neural network is well-known and has been used in previous studies i.e., PMID: 30866734, PMID: 36166351. Thus, the authors are suggested to refer to more works in this description to attract a broader readership.

Response: We have added to Section 5 the text: “We can find a lot of successful applications of neural networks to medical and biological tasks [104,105]. Neural networks also are used in the CATE estimation, for example, DRNet,…” and the following references:

  1. Nguyen Quoc Khanh Le and Quang-Thai Ho and Yu-Yen Ou. Using two-dimensional convolutional neural networks for identifying GTP binding sites in Rab proteins // Journal of Bioinformatics and Computational Biology. 17(01), 1950005 (2019)
  2. Quang-Hien Kha and Quang-Thai Ho and Nguyen Quoc Khanh Le. Identifying SNARE Proteins Using an Alignment-Free Method Based on Multiscan Convolutional Neural Network and PSSM Profiles // Journal of Chemical Information and Modeling. 2022, 62(19), 4820--4826.

Comment: The authors mentioned "The code of proposed algorithms implementing TNW-CATE is publicly available.", but we could not see it.

Response: It was a LaTeX problem. Site was working, but “-” in its title was not now included in the reference. The problem has been solved.

Reviewer 4 Report

The paper is interesting and represents a significant methodological advance in the relevant field of investigation. It is totally within the scope of the Journal and deserves attention. Overall, the presentation is quite clear and straight, and I do not have any major points to be raised. I just would like to know more about the efficiency of this approach in computational terms with respect to alternatives and, in the final part, a short discussion of possible use cases of this solution in everyday's research practice.

Author Response

Comment: I just would like to know more about the efficiency of this approach in computational terms with respect to alternatives and, in the final part, a short discussion of possible use cases of this solution in everyday's research practice.

Response: First, we have added to the Conclusion section that “The main limitation of the proposed model is the need to tune the hyperparameters of the neural network, which requires significant time spent on network training and validation. There are no strict heuristics that can significantly speed up this stage. However, this time is not crucial. For instance, the time of training on the dataset IHDP (see Subsection “Real dataset”)” is 1704 sec. whereas times of training the models T-RF, S-RF, X-RF are 49, 68k, 114 sec., respectively. With this time complexity, we pay for the accuracy of the results. In spite of the apparent complexity of the whole neural network for training, TNW-CATE is actually simple because it can be realized as a single small subnetwork implementing the kernel.”

Round 2

Reviewer 2 Report

.

Reviewer 3 Report

My previous comments have been addressed.